# Particle Probability Hypothesis Density Filter Based on Pairwise Markov Chains

**Jiangyi Liu [1], Chunping Wang [1,\*], Wei Wang [2] and Zheng Li [3]**

[1] Department of Electronic and optical engineering, Shijiazhuang Campus of Army Engineering University, Shijiazhuang 050000, China; liujiangyi054@gmail.com
[2] China Huayin Ordnance Test Center, Huayin 714200, China; wangwei1809@gmail.com
[3] Science and Technology on Aircraft Control Laboratory, Beihang University, Beijing 100000, China; lizheng@buaa.edu.cn
[\*] Correspondence: wchp054@gmail.com; Tel.: +86-27-87994237

**Abstract:** Most multi-target tracking filters assume that one target and its observation follow a Hidden Markov Chain (HMC) model, but the implicit independence assumption of the HMC model is invalid in many practical applications, and a Pairwise Markov Chain (PMC) model is more universally suitable than the traditional HMC model. A set of weighted particles is used to approximate the probability hypothesis density of multi-targets in the framework of the PMC model, and a particle probability hypothesis density filter based on the PMC model (PF-PMC-PHD) is proposed for the nonlinear multi-target tracking system. Simulation results show the effectiveness of the PF-PMC-PHD filter and that the tracking performance of the PF-PMC-PHD filter is superior to the particle PHD filter based on the HMC model in a scenario where we kept the local physical properties of nonlinear and Gaussian HMC models while relaxing their independence assumption.

**Keywords:** Pairwise Markov Chain; probability hypothesis density; particle filter; multi-target tracking system

## 1. Introduction

Random Finite Set (RFS) theory has been widely used in the multi-target tracking field. Unlike traditional solutions of multi-target tracking based on data association, RFS-based solutions provide a theoretical framework without data association [1,2]. Among RFS-based solutions, the Probability Hypothesis Density (PHD) filter propagates the first order moment of the posterior multi-target density [3], which is now widely applied. RFS-based solutions cannot get the analytical solution directly and the implementations mainly based on numerical approximations, such as Gauss Mixture (GM) PHD filter [4,5], and on Sequential Monte Carlo (SMC, i.e. particle filter) methods [6,7].

Most multi-target tracking filters, including the classical PHD filter, assume that the targets and the observations they produce follow the well-known HMC model. The HMC model assumes that the state of a given target is a Markov Chain (MC): the states of the current moment are determined only by the states of the previous moment, which have nothing to do with other moments, and the observations of the current moment is determined only by the states of the current moment. However, the Markovian and independence assumption implicit in the HMC model may not be satisfied in practical applications, such as situations of correlated process and measurement noises and situations of colored measurement noise [8]. In 2000, Pieczynski proposed the PMC model in order to relax the independence assumption of the HMC model [9–11]; the HMC model is a special PMC model and the PMC model is more universally suitable than HMC model. In 2013, Petetin and Desbouvries proposed a PHD filter for targets following the PMC model (PMC-PHD) [11] and proved that the tracking performance of the

proposed PMC-PHD filter is better than the "classical" PHD filter based on the HMC model under the situation which relaxed the independence assumptions implicit in the HMC model. The PMC-PHD filter proposed by Petetin only considers the first order information of the target state, neglecting its high order information, and leads to the instability of the target number estimation. In view of this problem, Mahler proposed a Cardinalized Probability Hypothesis Density filter based on the PMC model (PMC-CPHD), which is developed from the PMC-PHD filter by propagating the cardinality distribution function of the target simultaneously [12].

The GM implementation of the PMC-PHD filter proposed by Petetin and Desbouvries is only suitable for the linear Gaussian multi-target tracking system but not to a nonlinear system. GM implementations of the PHD filter use approximate methods to deal with non-linear problems, while particle implementations are not affected by linearity and non-linearity. In this paper, a set of weighted particles is used to approximate the probability hypothesis density of multi-targets in the framework of the PMC model, the probability hypothesis density of multi-targets is updated iteratively by predicting and updating the particles in real time to estimate the state of the targets, and a particle probability hypothesis density filter based on the PMC model (PF-PMC-PHD) is proposed for the nonlinear multi-target tracking system. The simulation result verifies the effectiveness of the PF-PMC-PHD filter and shows that the performance of PF-PMC-PHD is better than the particle implementation of the typical HMC-PHD filter (PF-HMC-PHD) in a scenario where we kept the local physical properties of nonlinear and Gaussian HMC models while relaxing their independence assumption.

The rest of the paper is organized as follows. A brief introduction to the PHD filter based on the PMC model is given in Section 2. The particle PMC-PHD (PF-PMC-PHD) is given for the nonlinear multi-target tracking system based on the PMC model in Section 3. The simulation results can be found in Section 4. The conclusions are in Section 5.

## 2. PHD Filter Based on the PMC Model

### 2.1. PMC Model

Let $x_k \in \mathbf{R}^m$ express the state at time $k$, and the corresponding observation is $y_k \in \mathbf{R}^q$. The couple $(x_k, y_k)$ is a PMC if and only if the joint probability density function (pdf) of $(x_{0:k}, y_{0:k})$ can be factorized as follows:

$$p(x_{0:k}, y_{0:k}) = p(x_0, y_0) \prod_{i=1}^{k} p_{i|i-1}(x_i, y_i | x_{i-1}, y_{i-1}) \tag{1}$$

where $p(x_0, y_0)$ is the state distribution at the initial time, and an HMC model is a PMC model which satisfies

$$\begin{cases} p(x_k | x_{k-1}, y_{k-1}) = p(x_k | x_{k-1}) = f(x_k | x_{k-1}) \\ p(y_k | x_k, x_{k-1}, y_{k-1}) = p(y_k | x_k) = g(y_k | x_k) \end{cases} \tag{2}$$

where $f(x_k | x_{k-1})$ and $g(y_k | x_k)$ are the target Markov transition density and the sensor likelihood function, respectively.

The defined target motion model follows the PMC model as

$$(x_k, y_k) = \varphi\big((x_{k-1}, y_{k-1}), w_k\big) \tag{3}$$

where $w_1, \cdots, w_k$ are independent zero-mean Gaussian noises, and

$$E(w_k w_k^T) = \Sigma_k = \begin{bmatrix} \Sigma_k^{11} & \Sigma_k^{21^T} \\ \Sigma_k^{21} & \Sigma_k^{22} \end{bmatrix} \tag{4}$$

A classical example of Gaussian PMC model can be expressed as [11]

$$
\begin{bmatrix} x_k \\ y_k \end{bmatrix} = \underbrace{\begin{bmatrix} F_k^1 & F_k^2 \\ H_k^1 & H_k^2 \end{bmatrix}}_{B_k} \begin{bmatrix} x_{k-1} \\ y_{k-1} \end{bmatrix} + w_k
\tag{5}
$$

### 2.2. PHD Filter Based on PMC Model

RFS-based solutions consider that at time $k$, targets and measurements are two RFS: $X_k = \{x_1, \cdots, x_n\}$ and $Z_k = \{z_1, \cdots, z_m\}$, where $n$ and $m$ are random integers which indicate the number of targets and measurements. The measurements associated to a given state $x_i$ is noted $y_i$ and defines the random finite set $\dot{X}_k = \{(x_1, y_1), \cdots, (x_n, y_n)\}$.

Representing PHD with $v(x)$, the $v(x)$ of the random finite set $X$ is the first order moment of multi-target density and also the spatial density of the expected number of targets. Directly computing $v_k(x)$ in a PMC seems complicated because $x_k$ is not necessarily Markovian; however, we can propagate a joint intensity $v_k(x, y)$ and then obtain the PHD $v_k(x)$ as $v_k(x) = \int v_k(x, y) \mathrm{d}y$.

Assuming that there is no spawning (if there is spawning the extension is immediate), according to the multi-target Bayesian principle, the joint PHD can be propagated through the following prediction and update formula:

$$
v_{k|k-1}(x, y) = \int p_{S,k}(x_{k-1}) f_{k|k-1}(x, y | x_{k-1}, y_{k-1}) \times v_{k-1}(x_{k-1}, y_{k-1}) dx_{k-1} dy_{k-1} + b_k(x, y)
\tag{6}
$$

$$
v_k(x, y) = [1 - p_{D,k}(x)] v_{k|k-1}(x, y) + \sum_{z \in Z_k} \frac{p_{D,k}(x) v_{k|k-1}(x, z) \delta_z(y)}{\kappa_k(z) + \int p_{D,k}(x) v_{k|k-1}(x, z) dx}
\tag{7}
$$

where $p_{S,k}(x)$ is the probability that a target with state $x$ at time $k-1$ still exists at time $k$; $p_{D,k}(x)$ is the probability that a target with state $x$ is detected at time $k$; $b_k(x, y)$ is the joint PHD of the birth targets RFS at time $k$; $\kappa_k(z)$ is the PHD of the clutter measurements RFS at time $k$; and $\delta_z(y)$ is the Dirac delta function concentrated at $z$. Finally remember that $v_k(x) = \int v_k(x, y) dy$, where the integral w.r.t. $y$ can reduce to a sum.

## 3. PF-PMC-PHD Filter

There are two main methods to implement a PHD filter, one is a particle implementation and the other is a GM implementation. The GM implementation of the PMC-PHD filter proposed by Petetin and Desbouvries is only suitable for a linear Gaussian multi-target tracking system. The particle PMC-PHD (PF-PMC-PHD) is given for the nonlinear multi-target tracking system based on the PMC model in this paper.

A set of weighted random samples $\left\{ w_k^{(i)}, (x_k^{(i)}, y_k^{(i)}) \right\}_{i=1}^{L}$ are used to approximate the posterior probability density function of the pair $(x, y)$ as follows:

$$
v_k(x, y) \approx \sum_{i=1}^{L} w_k^{(i)} \delta((x, y) - (x_k^{(i)}, y_k^{(i)}))
\tag{8}
$$

where $w_k^{(i)}$ represents the expected value of the pair with the state $(x_k^{(i)}, y_k^{(i)})$.

PF-PMC-PHD filter can be summarized as follow.

**Step 1:** Initialization of particles.

At time $k = 0$, use $L_0$ particles $\left\{ w_0^{(i)}, (x_0^{(i)}, y_0^{(i)}) \right\}_{i=1}^{L_0}$ to represent the prior probability density $v_0(\cdot)$ of pair $(x, y)$, and the particles number is proportional to the number of targets, that is, if there are $\hat{N}_0$ targets, $w_0^{(i)} = \hat{N}_0 / L_0$, the joint PHD function $v_0(x, y)$ of pair $(x, y)$ writes as

$$v_0(\boldsymbol{x}, \boldsymbol{y}) = \sum_{i=1}^{L_0} w_0^{(i)} \delta((\boldsymbol{x}, \boldsymbol{y}) - (\boldsymbol{x}_0^{(i)}, \boldsymbol{y}_0^{(i)})) \tag{9}$$

**Step 2:** Particles prediction.

At time $k \geq 1$, the particles $(\widetilde{\boldsymbol{x}}_k^{(i)}, \widetilde{\boldsymbol{y}}_k^{(i)})$ generating for the surviving targets are sampled from the proposed importance probability density function $q_k(\cdot | (\widetilde{\boldsymbol{x}}_{k-1}^{(i)}, \widetilde{\boldsymbol{y}}_{k-1}^{(i)}), Z_k)$, $i = 1, \cdots L_{k-1}$, and the particles $(\widetilde{\boldsymbol{x}}_k^{(i)}, \widetilde{\boldsymbol{y}}_k^{(i)})$, $i = L_{k-1} + 1, \cdots L_{k-1} + J_k$ generating for the new birth targets are sampled from another suggested density function $p_k(\cdot | Z_k)$. The corresponding predicted weights are calculated as

$$\widetilde{w}_{k|k-1}^{(i)} = \begin{cases} w_{k-1}^{(i)} \cdot \dfrac{\phi_{k|k-1}\left((\widetilde{\boldsymbol{x}}_k^{(i)}, \widetilde{\boldsymbol{y}}_k^{(i)}), (x_{k-1}^{(i)}, y_{k-1}^{(i)})\right)}{q_k((\widetilde{\boldsymbol{x}}_k^{(i)}, \widetilde{\boldsymbol{y}}_k^{(i)}) | (x_{k-1}^{(i)}, y_{k-1}^{(i)}), Z_k)} & i = 1, \cdots L_{k-1} \\[4mm] \dfrac{1}{J_k} \cdot \dfrac{b_k(\widetilde{\boldsymbol{x}}_k^{(i)}, \widetilde{\boldsymbol{y}}_k^{(i)})}{p_k((\widetilde{\boldsymbol{x}}_k^{(i)}, \widetilde{\boldsymbol{y}}_k^{(i)}) | Z_k)} & i = L_{k-1} + 1, \cdots L_{k-1} + J_k \end{cases} \tag{10}$$

Assuming that there is no spawning,

$$\phi_{k|k-1}\left((\widetilde{\boldsymbol{x}}_k^{(i)}, \widetilde{\boldsymbol{y}}_k^{(i)}), (x_{k-1}^{(i)}, y_{k-1}^{(i)})\right) = p_{S,k}(x_{k-1}^{(i)}) f_{k|k-1}(\widetilde{\boldsymbol{x}}_k^{(i)}, \widetilde{\boldsymbol{y}}_k^{(i)} | x_{k-1}^{(i)}, y_{k-1}^{(i)}) \tag{11}$$

The predicted joint PHD function $v_{k|k-1}(\boldsymbol{x}, \boldsymbol{y})$ of pair $(\boldsymbol{x}, \boldsymbol{y})$ writes as

$$v_{k|k-1}(\boldsymbol{x}, \boldsymbol{y}) = \sum_{i=1}^{L_{k-1}+J_k} \widetilde{w}_{k|k-1}^{(i)} \cdot \delta((\boldsymbol{x}, \boldsymbol{y}) - (\widetilde{\boldsymbol{x}}_k^{(i)}, \widetilde{\boldsymbol{y}}_k^{(i)})) \tag{12}$$

**Step 3:** Particles update.

Recalculating the weights of particles using measurements $z \in Z_k$ from the sensor, and the posterior probability density function $v_k(\boldsymbol{x}, \boldsymbol{y})$ of pair $(\boldsymbol{x}, \boldsymbol{y})$ writes as

$$v_k(\boldsymbol{x}, \boldsymbol{y}) = v_k^1(\boldsymbol{x}, \boldsymbol{y}) + v_k^2(\boldsymbol{x}, \boldsymbol{y}) \tag{13}$$

$$v_k^1(\boldsymbol{x}, \boldsymbol{y}) = \sum_{i=1}^{L_{k-1}+J_k} \widetilde{w}_k^{1,(i)} \delta((\boldsymbol{x}, \boldsymbol{y}) - (\widetilde{\boldsymbol{x}}_k^{(i)}, \widetilde{\boldsymbol{y}}_k^{(i)})) \tag{14}$$

$$\widetilde{w}_k^{1,(i)} = \left(1 - p_{D,k}(\widetilde{\boldsymbol{x}}_k^{(i)})\right) \widetilde{w}_{k|k-1}^{(i)} \tag{15}$$

$$v_k^2(\boldsymbol{x}, \boldsymbol{y}) = \sum_{z \in Z_k} \sum_{i=1}^{L_{k-1}+J_k} \widetilde{w}_k^{2,(i)}(z) \delta(\boldsymbol{x} - \widetilde{\boldsymbol{x}}_k^{(i)}) \delta_z(\boldsymbol{y}) \tag{16}$$

$$\widetilde{w}_k^{2,(i)}(z) = \dfrac{p_{D,k}(\widetilde{\boldsymbol{x}}_k^{(i)}) q_k^{(i)}(z) \cdot \widetilde{w}_{k|k-1}^{(i)}}{\kappa_k(z) + \sum\limits_{i=1}^{L_{k-1}+J_k} p_{D,k}(\widetilde{\boldsymbol{x}}_k^{(i)}) q_k^{(i)}(z) \widetilde{w}_{k|k-1}^{(i)}} \tag{17}$$

$$q_k^{(i)}(z) = \mathrm{N}(z; \widetilde{\boldsymbol{y}}_{k|k-1}^{(i)}; \Sigma_k^{22}) \tag{18}$$

**Step 4:** Resampling of the particles.

Estimating the number of targets:

$$\hat{N}_k = \sum_{i=1}^{L_{k-1}+J_k} \widetilde{w}_k^{1,(i)} + \sum_{z \in Z_k} \sum_{i=1}^{L_{k-1}+J_k} \widetilde{w}_k^{2,(i)}(z) \tag{19}$$

Resampling particles $\left\{ \widetilde{w}_k^{1,(i)} / \hat{N}_k, (\widetilde{\boldsymbol{x}}_k^{1,(i)}, \widetilde{\boldsymbol{y}}_k^{1,(i)}) \right\}_{i=1}^{L_{k-1}+J_k} \cup \left\{ \left\{ \widetilde{w}_k^{2,(i)}(\boldsymbol{z}_j) / \hat{N}_k, (\widetilde{\boldsymbol{x}}_k^{2,(i)}, \boldsymbol{z}_j) \right\}_{i=1}^{L_{k-1}+J_k} \right\}_{j=1}^{|Z_k|}$,
meanwhile, keeping the value $\boldsymbol{y}$ of particles which represent $v_k^2(\boldsymbol{x}, \boldsymbol{y})$, we will obtain particles $\left\{ w_k^{(i)} / \hat{N}_k, (\boldsymbol{x}_k^{(i)}, \boldsymbol{y}_k^{(i)}) \right\}_{i=1}^{L_k}$.

**Step 5:** Approximation of posterior probability density.

Rewrites the posterior joint PHD $v_k(\boldsymbol{x}, \boldsymbol{y})$ of pair $(\boldsymbol{x}, \boldsymbol{y})$ as

$$v_k(\boldsymbol{x}, \boldsymbol{y}) = \sum_{i=1}^{L_k} w_k^{(i)} \delta((\boldsymbol{x}, \boldsymbol{y}) - (\boldsymbol{x}_k^{(i)}, \boldsymbol{y}_k^{(i)})) \tag{20}$$

According to $v_k(\boldsymbol{x}) = \int v_k(\boldsymbol{x}, \boldsymbol{y}) d\boldsymbol{y}$, and $\int \delta(\boldsymbol{y} - \boldsymbol{y}_k^{(i)}) d\boldsymbol{y} = 1$, $i = 1, \cdots L_{k-1} + J_k$, the posterior PHD $v_k(\boldsymbol{x})$ of state $\boldsymbol{x}$ writes as

$$v_k(\boldsymbol{x}) = \sum_{i=1}^{L_k} w_k^{(i)} \delta(\boldsymbol{x} - \boldsymbol{x}_k^{(i)}) \tag{21}$$

## 4. Experimental Simulation

### 4.1. A Particular Class of Gaussian PMC Model

In order to verify the effectiveness of the proposed PF-PMC-PHD filter and compare the tracking performance with the particle PHD filter based on the HMC model (PF-HMC-PHD), the experimental simulation uses a special Gaussian PMC model, with $p(\boldsymbol{x}_k|\boldsymbol{x}_{k-1})$ and $p(\boldsymbol{y}_k|\boldsymbol{x}_k)$ that are the same as $f_{k|k-1}(\boldsymbol{x}_k|\boldsymbol{x}_{k-1})$ and $g_k(\boldsymbol{y}_k|\boldsymbol{x}_k)$ of the HMC model while relaxing the independence assumption. The two models have the same local physical properties, but remember that in general $p(\boldsymbol{x}_k|\boldsymbol{x}_{k-1})$ will not be equal to $p(\boldsymbol{x}_k|\boldsymbol{x}_{k-1}, \boldsymbol{y}_{k-1})$ and $p(\boldsymbol{y}_k|\boldsymbol{x}_k)$ will not be equal to $p(\boldsymbol{y}_k|\boldsymbol{x}_k, \boldsymbol{x}_{k-1}, \boldsymbol{y}_{k-1})$ in the PMC model, while the HMC model satisfies Equation (2). Suppose that for all $k$, the HMC model satisfies

$$p(\boldsymbol{x}_0) = N(\boldsymbol{x}_0; \boldsymbol{m}_0, P_0) \tag{22}$$

$$f_{k|k-1}(\boldsymbol{x}_k|\boldsymbol{x}_{k-1}) = N(\boldsymbol{x}_k; F_k \boldsymbol{x}_{k-1}, Q_k) \tag{23}$$

$$g_k(\boldsymbol{y}_k|\boldsymbol{x}_k) = N(\boldsymbol{y}_k; H_k \boldsymbol{x}_k, R_k) \tag{24}$$

The corresponding Gaussian PMC model can be expressed by the following formula [11]:

$$p(\boldsymbol{\xi}_0) = N\left( \boldsymbol{\xi}_0; \begin{bmatrix} \boldsymbol{m}_0 \\ H_0 \boldsymbol{m}_0 \end{bmatrix}, \begin{bmatrix} P_0 & (H_0 P_0)^{\mathrm{T}} \\ H_0 P_0 & R_0 + H_0 P_0 H_0^{\mathrm{T}} \end{bmatrix} \right) \tag{25}$$

$$p_{k|k-1}(\boldsymbol{\xi}_k|\boldsymbol{\xi}_{k-1}) = N(\boldsymbol{\xi}_k; B_k \boldsymbol{\xi}_{k-1}, \Sigma_k) \tag{26}$$

where $\boldsymbol{\xi}_k = (\boldsymbol{x}_k, \boldsymbol{y}_k)$ and

$$B_k = \begin{bmatrix} F_k - F_k^2 H_{k-1} & F_k^2 \\ H_k F_k - H_k^2 H_{k-1} & H_k^2 \end{bmatrix}, \Sigma_k = \begin{bmatrix} \Sigma_k^{11} & \Sigma_k^{21^T} \\ \Sigma_k^{21} & \Sigma_k^{22} \end{bmatrix} \tag{27}$$

$$\Sigma_k^{11} = Q_k - F_k^2 R_{k-1}(F_k^2)^{\mathrm{T}} \tag{28}$$

$$\Sigma_k^{21} = H_k Q_k - H_k^2 R_{k-1}(F_k^2)^{\mathrm{T}} \tag{29}$$

$$\Sigma_k^{22} = R_k - H_k^2 R_{k-1}(H_k^2)^{\mathrm{T}} + H_k Q_k(H_k)^{\mathrm{T}} \tag{30}$$

### 4.2. Performance Analysis

Let us now analyze quantitatively the tracking performance of the PF-PMC-PHD filter and PF-HMC-PHD filter in a nonlinear system based on the PMC model. We compute at each time step the OSPA (Optimal Sub-Patten Assignment) metric and the target number estimation. The OSPA metric can simultaneously evaluate the target number estimation error and position error of the multi-target tracking. Let $X = \{x_1, \cdots, x_m\}$ and $\hat{X} = \{\hat{x}_1, \cdots, \hat{x}_n\}$ be two finite sets. Here, $\hat{X}$ represents the estimated finite set of the targets and $X$ represents the true finite set of targets. For $1 \leq p < +\infty$ and $c > 0$, let $d^{(c)}(x, \hat{x}) = \min(c, \|x - \hat{x}\|)$ ($\|\cdot\|$ is the Euclidean norm), $\Pi_n$ be the set of permutations on $\{1, 2, \cdots, n\}$, and let $\pi(i)$ be the $i$th component of a given permutation $\pi$. The OSPA metric is defined by [13]

$$\overline{d}_p^c(X, \hat{X}) \triangleq \left( \frac{1}{n} \left( \min_{\pi \in \Pi_n} \sum_{i=1}^{m} d^{(c)}(x_i, \hat{x}_{\pi(i)})^p + c^p(n - m) \right) \right)^{\frac{1}{p}} \tag{31}$$

if $m \leq n$ and by $\overline{d}_p^c(X, \hat{X}) \triangleq \overline{d}_p^c(\hat{X}, X)$ if $m > n$. In our simulations, we set $p = 100$ and $c = 1$. Set

$$F_k = \begin{bmatrix} 1 & \frac{\sin \Omega T}{\Omega} & 0 & -\frac{1 - \cos \Omega T}{\Omega} \\ 0 & \cos \Omega T & 0 & -\sin \Omega T \\ 0 & \frac{1 - \cos \Omega T}{\Omega} & 1 & \frac{\sin \Omega T}{\Omega} \\ 0 & \sin \Omega T & 0 & \cos \Omega T \end{bmatrix}, F_k^2 = \begin{bmatrix} 0.7 & 0 \\ 0 & 0 \\ 0 & 0.7 \\ 0 & 0 \end{bmatrix} \tag{32}$$

$$H_k = \begin{bmatrix} 1 & 0 & 0 & 0 \\ 0 & 0 & 1 & 0 \end{bmatrix}, H_k^2 = \begin{bmatrix} 0.1 & 0 \\ 0 & 0.1 \end{bmatrix} \tag{33}$$

$$Q_k = \begin{bmatrix} 25 & 1 & 0 & 0 \\ 1 & 5 & 0 & 0 \\ 0 & 0 & 25 & 1 \\ 0 & 0 & 1 & 5 \end{bmatrix}, R_k = \begin{bmatrix} 4 & 0 \\ 0 & 4 \end{bmatrix} \tag{34}$$

We generate uniformly a mean of 10 clutter measurements on the region $V = [-2000, 2000] \times [-2000, 2000]$ with a sampling period of $T = 1s$ and a simulation experiment step of $N = 50$. There are 4 targets: targets 1 and 2 appear at time $k = 1$ and targets 3 and 4 appear at time $k = 20$. We track the position and velocity of the targets in Cartesian coordinates, $\mathbf{x}_k = \begin{bmatrix} p_{\mathbf{x},k}, \dot{p}_{\mathbf{x},k}, p_{\mathbf{y},k}, \dot{p}_{\mathbf{y},k} \end{bmatrix}^T$. We set $p_{S,k} = 0.98$ and $p_{D,k} = 0.9$ and select $L = 1000$ for the particle number of one target. The target trajectories and measurements of such a scenario is displayed in Figure 1.

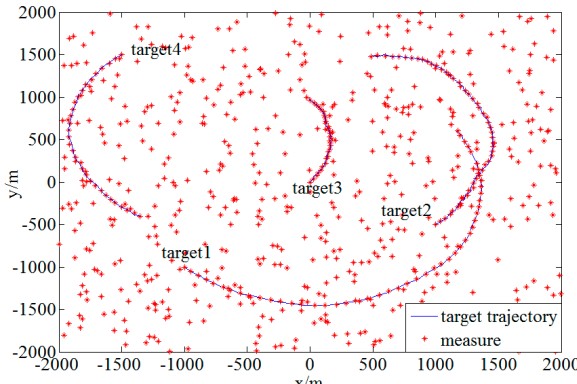

**Figure 1.** The target trajectories and measurements: Trajectories of four targets are represented by lines and the measurements taken from the sensor are represented by "Union Jack".

Figure 2 shows the tracking result of PF-PMC-PHD. The tracking result shows that the proposed PF-PMC-PHD filter can effectively achieve the target tracking.

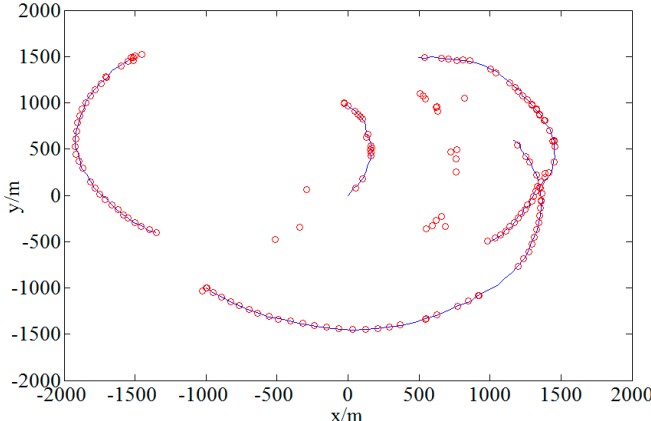

**Figure 2.** The tracking result of PF-PMC-PHD: Target trajectories are represented by lines, and the tracking results are represented by circles. The algorithm can track the targets effectively at most times, but there are missed and false alarms at some times.

Figures 3 and 4 respectively show target number estimations and the OSPA metrics of the PF-PMC-PHD filter and PF-HMC-PHD filter. The simulation result shows that the tracking performance of the PF-PMC-PHD filter is better than that of PF-HMC-PHD filter although the two filters share the same $p(x_k|x_{k-1})$ and $p(y_k|x_k)$. This is because the HMC model does not take into account the information given by the observation $y_{k-1}$ in the case of the given $x_{k-1}$ and $y_{k-1}$ and because the uncertainty of the state $x_k$ increases. The target number estimation error of the PF-HMC-PHD filter is large, and the situation of missing target is serious.

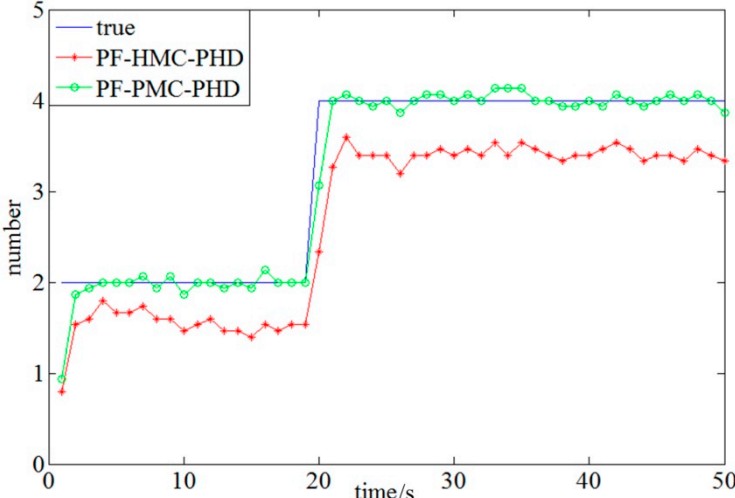

**Figure 3.** The target number estimations and PF-HMC-PHD (L = 1000): The number estimation of PF-PMC-PHD is approximate to the true number when L = 1000, while the number estimation of PF-HMC-PHD has a large error with the true number.

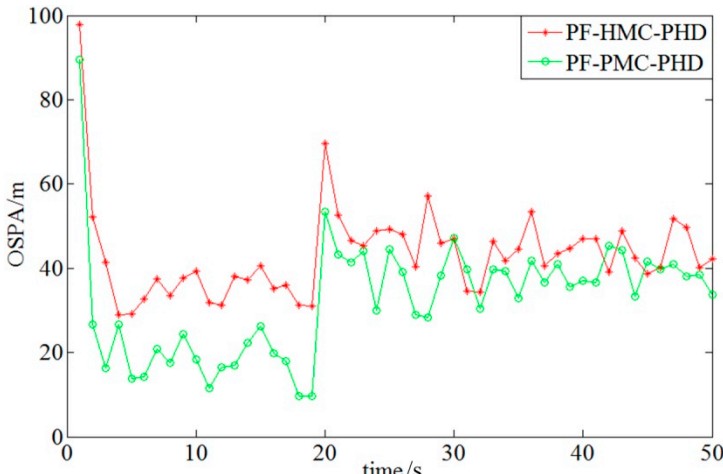

**Figure 4.** The OSPA metrics of PF-PMC-PHD and PF-HMC-PHD (L = 1000): The OSPA distance of PF-PMC-PHD is smaller to that of PF-HMC-PHD.

In order to verify the effect of the particle number for each target on the tracking performance of PF-PMC-PHD filtering and PF-HMC-PHD filtering, change the value of L. Figures 5 and 6 respectively show the target number estimations and the OSPA metrics of the PF-PMC-PHD filter and PF-HMC-PHD filter when L = 500, and Figures 7 and 8 respectively show the target number estimations and the OSPA metrics of the PF-PMC-PHD filter and PF-HMC-PHD filter when L = 2000. The tracking performance of the PF-PMC-PHD filter and PF-HMC-PHD filter improve gradually with the increase of the L value. This is because with the increase of the particle number, the particle PHD filter is more close to the real PHD.

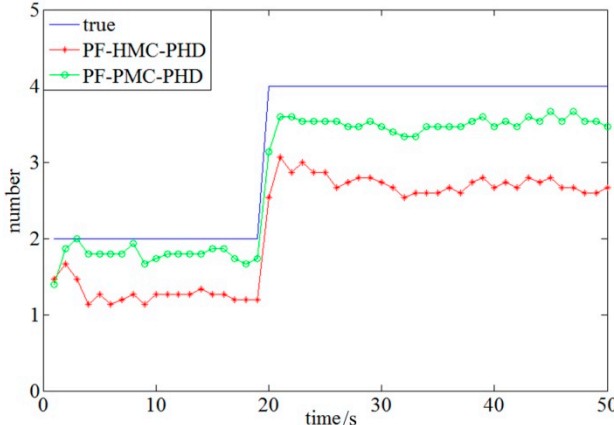

**Figure 5.** The target number estimations of PF-PMC-PHD and PF-HMC-PHD (L = 500): The target number estimations of the two filters are smaller than the true number when L = 500, and the error of PF-HMC-PHD is larger than that of PF-PMC-PHD.

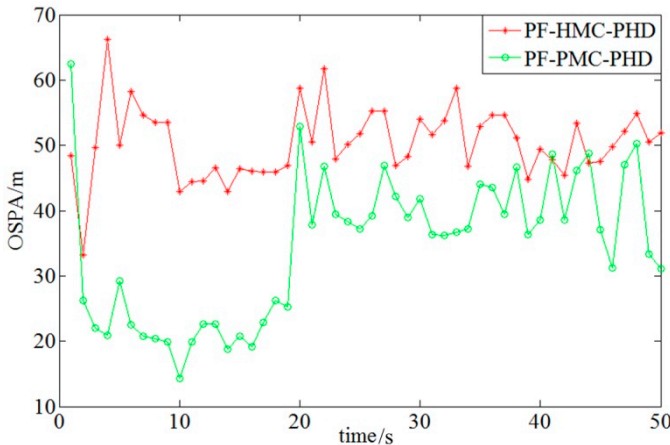

**Figure 6.** The SPA metrics of PF-PMC-PHD and PF-HMC-PHD (L = 500). The OSPA distance of PF-PMC-PHD is smaller than that of PF-HMC-PHD.

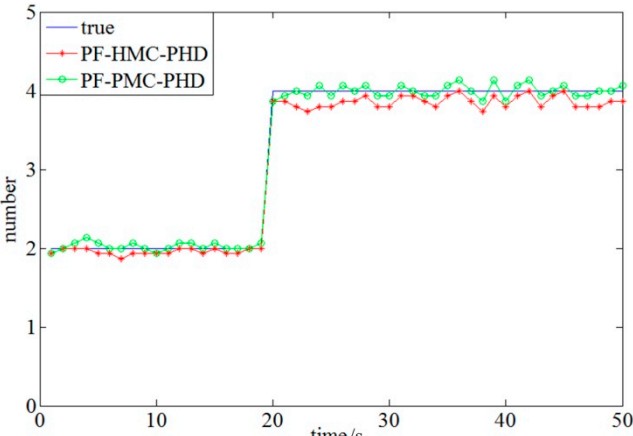

**Figure 7.** The target number estimations of PF-PMC-PHD and PF-HMC-PHD (L = 2000): The number estimations of the two filters are approximate to the true number when L = 2000, and the error of PF-HMC-PHD is larger than that of PF-PMC-PHD.

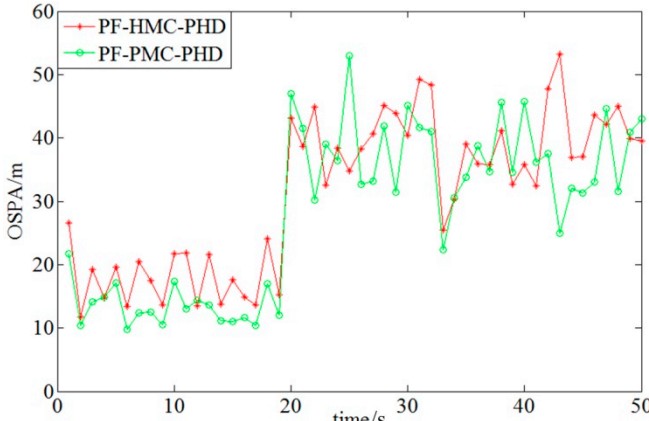

**Figure 8.** The OSPA metrics of PF-PMC-PHD and PF-HMC-PHD (L = 2000): The OSPA distance of the two filters are similar when L = 2000.

## 5. Conclusions

The Pairwise Markov Chain (PMC) model is more universally suitable than the traditional Hidden Markov Chain (HMC) model. A PF-PMC-PHD filter is proposed for a nonlinear multi-target tracking system based on the PMC model. The simulation result proves the effectiveness of the filter when the

particle number for one target is more than 1000, and the tracking accuracy needs to be improved when the particles number for one target is small. The simulation result also shows that the performance of the PF-PMC-PHD filter is better than the traditional PF-HMC-PHD filter in a scenario where we kept the local physical properties of the nonlinear and Gaussian HMC models while relaxing their independence assumption.

In order to verify the superiority of the PF-PMC-PHD filter over PF-HMC-PHD filter, this paper adopts a certain and known PMC model. But in practical applications, the PMC model is generally unknown and uncertain, and further studies in these complex situations are needed [14,15].

**Author Contributions:** Conceptualization, J.L. and W.W.; methodology, J.L.; software, J.L.; formal analysis, J.L.; investigation, J.L.; resources, J.L.; data curation, J.L.; writing—original draft preparation, J.L.; writing—review and editing, J.L. and Z.L.; supervision, C.W. and W.W.; project administration, J.L.; funding acquisition, C.W.

**Funding:** This research was funded by Natural Science Foundation of China, grant number 61141009.

**Conflicts of Interest:** The authors declare no conflict of interest.

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
