# Peer review of "Particle Probability Hypothesis Density Filter Based on Pairwise Markov Chains"

_algorithms, doi:10.3390/a12020031_

Round 1
Reviewer 1 Report
The paper is interesting and contains interesting material, in my opinion. Generally, It is also well-written and well-structured. However, I think that several points must be improved before publication.
My main concern is the clarity and the possible impact of this work. I have some suggestions to improve the paper and its impact. See below.
- please, explain better your contribution in the introduction.
- please, describe the structure of the paper in a last paragraph before of Section 2.
- At line 58, page 2, \mathbb{R} is not visible (they seems rectangles). is it ok?
- please, explain what is v(x) in Section 2.2, line 77. Improve its description.
- Please, improve the caption of Figures 1, 2, 3, 4, 5, 6, 7 and 8 (giving more explanation).
- It is important to improve the state-of-the-art to increase the impact and appealing of the paper. For instance, in my opinion, it is very important to discuss similar applications to the multi-target tracking. For instance the model selection problem is solved by using similar schemes (at least, when the number of models is known). I suggest to add references to benchmark and modern techniques related to model selection, for instance:
- I. Urteaga, M. F. Bugallo, and P. M. Djuric. Sequential Monte Carlo methods under model uncertainty, IEEE Statistical Signal Processing Workshop (SSP), pages 15, 2016.
- L. Martino, J. Read, V. Elvira, F. Louzada, Cooperative Parallel Particle Filters for on-Line Model Selection and Applications to Urban Mobility, Digital Signal Processing Vol. 60, pp. 172-185, 2017.
- C. C. Drovandi, J. McGree, and A. N. Pettitt. A sequential Monte Carlo algorithm to incorporate model uncertainty
in Bayesian sequential design." Journal of Computational and Graphical Statistics, 23(1):324, 2014.
It can increase the number of interested readers of your work (increasing your audience).
- Please, upload the final version of your manuscript in Arxiv and/or ResearchGate when/if published, to increase the diffusion and the possible citations of your work.
Author Response
A point-by-point response to the reviewer’s comments is upload as a Word file

Reviewer 2 Report
Paper can be accepted after the following corrections:
Possible practical applications should be clearly stated in the introduction. Criteria of practical usability for tracking filter should be determined.
Line 58 should be corrected.
Figure caption of figure 1 should be developed to be self-explaining.
Figures 3 - 8: Please define units for "time"
OSPA acronym should be explained.
Conclusions should be developed and clearly stated in quantitative way.
Author Response
A point-by-point response to the reviewer’s comments is upload it as a Word file

Reviewer 3 Report
See attached file

Author Response

(The authors gave the same response as above.)

Reviewer 4 Report
The authors present a particle filter to implement a PMC-PHD filter for non-linear multi-target tracking. This is a real contribution with respect to the initial work by petetin/desbouvries that were first to combine PMC and PHD. While interesting, I found difficult to follow Section 2. For example:
- the if sentence in line 58-65 is difficult to understand. What is the 'then ' corresponding to the 'if'?
- if I understand well paragraph starting by line 73, Z_k i a measurment. Do you mean Y_k?
- line 90 : 'y can reduce to a sum'. What do you mean? Are you talking about the approximation of a continuous integral by a sum? If yes, what is the value used in experiments? What is the impact of the number of discrete steps on the results.
Also it seems that there is problem in caption of figures 5 to 8.
Can you define OSPA error metric?
Author Response

(The authors gave the same response as above.)

Round 2
Reviewer 3 Report
My recommendations have been correctly taken into account.
This manuscript is a resubmission of an earlier submission. The following is a list of the peer review reports and author responses from that submission.